# Breaking the MoE LLM Trilemma: Dynamic Expert Clustering with Structured Compression

## Abstract

Mixture-of-Experts (MoE) Large Language Models (LLMs) face a trilemma of load imbalance, parameter redundancy, and communication overhead. We introduce a unified framework based on dynamic expert clustering and structured compression to address these issues cohesively. Our method employs an online clustering procedure that periodically regroups experts using a fused metric of parameter and activation similarity, which stabilizes expert utilization. To our knowledge, this is one of the first frameworks to leverage the semantic embedding capability of the router to dynamically reconfigure the model's architecture during training for substantial efficiency gains. Within each cluster, we decompose expert weights into a shared base matrix and extremely low-rank residual adapters, achieving up to fivefold parameter reduction per group while preserving specialization. This structure enables a two-stage hierarchical routing strategy: tokens are first assigned to a cluster, then to specific experts within it, drastically reducing the routing search space and the volume of all-to-all communication. Furthermore, a heterogeneous precision scheme, which stores shared bases in FP16 and residual factors in INT4, coupled with dynamic offloading of inactive clusters, reduces peak memory consumption to levels comparable to dense models. Evaluated on GLUE and WikiText-103, our framework matches the quality of standard MoE models while reducing total parameters by approximately 80%, improving throughput by 10% to 20%, and lowering expert load variance by a factor of over three. Our work demonstrates that structural reorganization is a principled path toward scalable, efficient, and memory-effective MoE LLMs. Code for experiments is available at https://anonymous.4open.science/r/SUBMIT-0001/README.md

## 1 Introduction

Mixture-of-Experts (MoE) architectures have become a key technical path for scaling large language models (LLMs) (Shazeer et al., 2017; Lepikhin et al., 2020; Fedus et al., 2021; Du & et al., 2021; Jiang et al., 2024; Rajbhandari et al., 2022), offering a path to exponential model growth (Lepikhin et al., 2020; Fedus et al., 2021) without a proportional surge in computational costs. By routing inputs to specialized "expert" subnetworks, MoE theoretically enables efficient and dynamic resource allocation. However, deploying these models on modern hardware like GPUs reveals a fundamental "optimization trilemma": The three critical bottlenecks of load imbalance, parameter redundancy, and communication overhead each present significant challenges on their own. Beyond their individual difficulties, these three bottlenecks are also deeply intertwined and mutually constraining, forming the core of the fundamental "optimization trilemma" that hinders MoE model efficiency.

This trilemma is not merely theoretical but a direct consequence of hardware and system limitations. The vast parameter counts of MoE models strain limited GPU memory capacity, making redundancy a critical issue. Load imbalance leads to underutilization of expensive compute units, diminishing throughput gains. Most critically, the all-to-all communication (Shazeer et al., 2017; Lepikhin et al., 2020) required to route tokens between experts across different devices often becomes the dominant latency bottleneck, especially with long sequences (Rajbhandari et al., 2022; Hwang et al., 2023).

Existing optimization methods tend to address these issues in isolation, leading to fragmented and often counterproductive solutions. For instance, load-balancing losses (Shazeer et al., 2017; Fedus et al., 2021) are reactive and struggle with distribution shifts. Parameter compression techniques like pruning or quantization, as seen in MoE-Lite (Kossai et al., 2023), reduce memory but treat experts as independent entities, ignoring potential structured similarities. Communication-aware routing (Hwang et al., 2023; He et al., 2022) optimizes data transfer for a fixed architecture but cannot address the underlying parameter redundancy or imbalance (Hubara et al., 2018; Jacob et al., 2018). This siloed approach highlights a key contradiction: Efforts to solve one bottleneck often exacerbate another. A unified framework to resolve these three intrinsic conflicts simultaneously is desperately lacking.

We argue that the key to breaking this vicious cycle lies in fundamentally reshaping expert organization via a dynamic, structured hierarchy, which rooted in our core insight that experts activated by semantically similar inputs also exhibit parameter redundancy, enabling their dynamic grouping. This allows us to co-design the architecture and system optimizations. Our framework integrates four tightly-coupled innovations:

1. **Online Expert Clustering for Dynamic Load Balancing:** We propose an online clustering algorithm, its core is partitioning experts into groups periodically based on a dual similarity metric (parameter similarity and activation similarity). This algorithm stabilizes the utilization of expert groups and forms cohesive expert groups.

2. **Intragroup Parameter Compression via Low-Rank Residuals:** For each expert group, we decompose each expert's weight into a shared base matrix and a low-rank residual adapter. By leveraging intra-group similarity, this decomposition drastically reduces parameters while avoiding the loss of model expressive power.

3. **Hierarchical Routing for Communication Efficiency:** We design a two-stage routing process (first assigning tokens to groups, then to specific experts within groups) based on the expert group structure. This process significantly reduces the routing search space and the volume of all-to-all communication.

4. **Dynamic Offloading and Quantization for Memory Optimization:** We adopt two strategies for memory optimization: A heterogeneous precision scheme (FP16 bases, INT4 residuals) and dynamic offloading of inactive expert groups. Together, these strategies reduce peak memory consumption to levels comparable to dense models.

Our goal is to design a dynamic grouping and parameterization that (i) reduces total stored parameters, (ii) keeps per-token activated parameters low, (iii) improves or maintains task quality, and (iv) reduces inter-device traffic without incurring large reclustering overhead. Our experiments show this unified approach reduces total parameters by approximately 80% and improves throughput by 10% to 20% compared to standard MoE models, while matching their quality.

## 2 RELATED WORK

### 2.1 EXPERT DESIGN AND PARAMETER EFFICIENCY

The vast parameter count of MoE models, primarily from replicating expert weights, has spurred research into more efficient expert designs. Traditional approaches, such as pruning and quantization (Han et al., 2015; Hubara et al., 2018; Jacob et al., 2018) as seen in MoE-Lite (Kossai et al., 2023), treat experts as independent entities to be compressed. A more profound line of inquiry questions the nature of an "Expert". Mixture-of-Lookup-Experts (MoLE) (Jie et al., 2025) radically replaces MLPs with parameter-free lookup tables, sacrificing expressive power for efficiency. Closer to our work, some methods merge similar experts. Expert-Fusion does so during training, risking a permanent loss of specialization, while Sub-MoE (Li et al., 2025) leverages subspace alignment to resolve parameter conflicts during merging. For fine-tuning, PERFT (Liu et al., 2024) integrates PEFT modules with MoE routing. Our approach differs by using dynamic clustering to enable parameter sharing via low-rank residuals, which preserves specialization while achieving high compression without permanent merging.

## 2.2 ROUTING, LOAD BALANCING, AND SEMANTIC SPECIALIZATION

The router is the brain of an MoE. Early work focused on mitigating load imbalance through auxiliary losses (Shazeer et al., 2017; Fedus et al., 2021), a reactive approach. A conceptual leap came from the insight that a router's output logits are powerful semantic embeddings of input tokens (Li & Zhou, 2024). This reframes the router from a simple gate to a rich feature extractor and provides a theoretical foundation for similarity-based expert organization. While this finding was used for analysis, our method is one of the first to harness this semantic space to dynamically reconfigure the model's architecture during training for tangible efficiency gains. Other works have focused on computational load and stability. MoE++ (Jin et al., 2024) introduced a "zero-computation" expert to skip easy tokens, while Huang et al. (2024) proposed dynamically adjusting the number of activated experts. StableMoE (Dai et al., 2022) uses a two-stage training to stabilize routing. Our hierarchical routing builds on these ideas but is structurally different: by routing to groups first, we inherently reduce the search space and stabilize load at a coarser level before fine-grained assignment.

## 2.3 SYSTEM-LEVEL OPTIMIZATIONS

Ultimately, an MoE model's success is measured by its end-to-end performance. Recent open-source models like OLMoE (Muennighoff et al., 2024) have demonstrated state-of-the-art performance by co-designing training recipes and architectures. However, many still rely on a static MoE structure, tackling system challenges through expert parallelism and sophisticated communication libraries. For instance, Tutel (Hwang et al., 2023), MoE-Lightning (Cao et al., 2025), and SmILE (He et al., 2022) introduce topology-aware routing to optimize communication paths but operate on a fixed model architecture. HOBBIT (Tang et al., 2024) dynamically manages GPU memory by caching experts.

## 2.4 HARDWARE CONSTRAINTS OF MoE DEPLOYMENT

The practical deployment of MoE models on modern accelerators (e.g., NVIDIA A100/H100 GPUs) exacerbates the trilemma due to inherent hardware constraints. Key limitations include such as Memory Bandwidth and Capacity: The vast parameter count from numerous experts strains limited high-bandwidth memory (HBM) capacity, making parameter redundancy a critical bottleneck. Compute Utilization: Load imbalance leads to poor saturation of compute units, diminishing theoretical throughput gains (Hwang et al., 2023). And Interconnect Bandwidth: The all-to-all communication pattern (Shazeer et al., 2017; Lepikhin et al., 2020) for token routing consumes significant inter-device interconnect bandwidth (e.g., NVLink, InfiniBand), often becoming the dominant source of latency. Although on-chip SRAM bandwidth is high, frequent inter-device exchanges and irregular memory access patterns lead to memory-bound stalls (Williams et al., 2009). These observations motivate a co-design of the MoE architecture and the underlying system, moving beyond localized optimizations (Choquette, 2023).

In summary, prior work has made significant strides through isolated improvements. Compression techniques often ignore load balance, dynamic routing can introduce instability, and system optimizations are typically applied to static architectures.

## 3 METHOD

To address the remaining issues from previous studies, our method introduces a unified framework to break the MoE trilemma through dynamic expert clustering and structured parameter compression. The core idea is to co-optimize the model's architecture alongside its parameters. We frame this as a optimization problem, formally defined as below:

$$\min_{Grouping,\ Param,\ Factors,\ Routing} L_{\text{task}} + A_1 \underbrace{I_{\text{load}}}_{imbalance} + A_2 \underbrace{R_{\text{red}}}_{redundancy} + A_3 \underbrace{C_{\text{comm}}}_{dispatch\,cost} \qquad (1)$$

The optimization variables correspond to three key aspects: the assignment of experts into cohesive clusters ($Grouping$), the compressed parameterization of experts via shared bases and low-rank residuals ($Param$ and $Factors$), and the policy for hierarchically routing tokens ($Routing$). The hyperparameters $A_1$, $A_2$, $A_3$ balance the relative importance of the load imbalance, redundancy,

and communication cost terms respectively. The objectives include the task-specific loss $L_{\text{task}}$, the coefficient of variation of expert loads $I_{\text{load}}$ (quantifying load imbalance), a measure of parameter redundancy $R_{\text{red}}$ within clusters, and the communication cost $C_{\text{comm}}$ incurred by token dispatch. The overall aim is to design a dynamic grouping and parameterization strategy that reduces the total parameter count, maintains a low number of activated parameters per token, preserves or improves model quality, and reduces inter-device communication, all without introducing significant reclustering overhead.

### 3.1 ONLINE DUAL-SIMILARITY CLUSTERING

The standard $Top - k$ (where each token is routed to the $k$ most relevant experts) gating mechanism over a large set of $E$ experts often leads to load imbalance when the input token distribution shifts, causing some experts to saturate while others remain underutilized. To address this, we propose to dynamically reorganize the experts themselves, rather than solely adjusting the routing probabilities.

Our clustering is based on a fused similarity metric that captures both structural and functional characteristics of experts.While complementary work (Li et al., 2025) validates data-driven clustering for experts, we extend it by incorporating both parameter and activation similarities. Each expert $\mathcal{E}_i$ is represented by two features: Its parameter vector, obtained by flattening its weight matrix $W_i$, and an activation centroid $\mu_i$. This centroid $\mu_i$, is an exponentially decayed mean of token embeddings routed to that expert. Based on these, we define parameter similarity and task similarity scores. The parameter similarity $S_{\text{param}}$, is the cosine similarity between the experts' weight vectors:

$$S_{\text{param}}(\mathcal{E}_i, \mathcal{E}_j) = \frac{\langle vec(W_i), vec(W_j) \rangle}{\|vec(W_i)\| \|vec(W_j)\|} \tag{2}$$

The task/activation similarity $S_{\text{task}}$, is the cosine similarity between their activation centroids:

$$S_{\text{task}}(\mathcal{E}_i, \mathcal{E}_j) = \frac{\mu_i^\top \mu_j}{\|\mu_i\| \|\mu_j\|} \tag{3}$$

These two metrics are combined into a single fused similarity score $S$, using a fusion weight $\alpha$ (where $\alpha \in [0, 1]$ controls the relative importance between parameter and task similarity). We usually consider that when the value of $\alpha$ is greater than 0.5, it by default places greater emphasis on parameter similarity, because parameters are the direct embodiment of functions. The formula for the fused similarity score is:

$$S(\mathcal{E}_i, \mathcal{E}_j) = \alpha S_{\text{param}}(\mathcal{E}_i, \mathcal{E}_j) + (1 - \alpha) S_{\text{task}}(\mathcal{E}_i, \mathcal{E}_j), \quad \alpha \in [0, 1] \tag{4}$$

The centroids $\mu_i$ are updated after each routing step via Exponential Moving Average (EMA), a smoothing technique governed by an update rate $\beta$ (default 0.05):

$$\mu_i \leftarrow (1 - \beta)\mu_i + \beta \bar{x}_i \tag{5}$$

where $\bar{x}_i$ is the mean embedding of tokens assigned to expert $i$ in the current step, if no tokens arrive, $\mu_i$ remains unchanged, if there are $N$ tokens are allocated, $\overline{x}_i = \frac{1}{N} \sum_{t=1}^{N} x_t$. The clustering procedure is performed periodically every $T$ training steps. The procedure is as follows: (i) optionally, we construct an approximate neighbor graph by discarding expert pairs with a similarity score below a threshold $\tau$ (default 0.1), which helps to limit the $O(E^2)$ comparisons, (ii) we run K-means++ seeding on the distance metric $D = 1 - S$ to produce $G$ groups, each of a uniform size $K = E/G$. If slight imbalance emerges, we greedily rebalance by moving boundary experts, which mean experts with high affinity to a different cluster, with minimal similarity loss (Arthur & Vassilvitskii, 2007). Group assignments are then sent to all workers. To amortize cost, we cache the parameter similarity matrix $S_{\text{param}}$ for a duration of $m$ steps (the cache lifetime), recomputing it only for experts whose weights have changed beyond a relative update norm of $\epsilon$. This dynamic grouping yields several key benefits: (i) enforces high intragroup correlation, enabling accurate low-rank residualization, (ii) reduces first-stage routing complexity from $O(E)$ to $O(G)$ , (iii) smooths token skew because group-level dispatch acts as a coarse pre-balancer, (iv) enables group-aware memory policies (Sect. 3.4).

### 3.2 SHARED BASE WITH LOW-RANK RESIDUALS

We visualize the clustering and compression process in Figure 2. Experts in the same group exhibit correlated transformations. Storing all $K$ full-sized weight matrices, each of dimension $d_{in} \times d_{out}$,

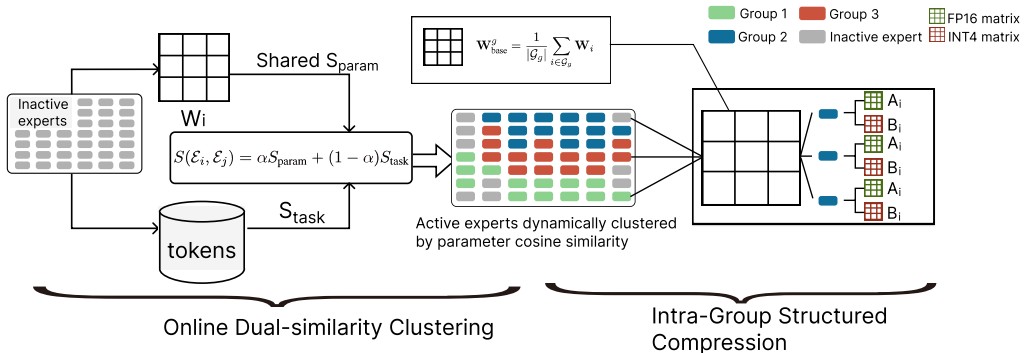

Figure 1: Overview of the online dual-similarity clustering and intra-group structured compression. Experts are dynamically clustered based on a fused similarity metric. Within each group, experts are compressed into a shared base matrix and low-rank residual adapters

is therefore wasteful. We exploit this by decomposing each expert's weight matrix $W_i$ into a shared base and a compact, expert-specific residual. Formally, for any group $g$ with a set of expert indexes $\mathcal{G}_g$, we first compute a shared base matrix $W_{\text{base}}^g$, by averaging the weights of all experts within that group:

$$W_{\text{base}}^g = \frac{1}{|\mathcal{G}_g|} \sum_{i \in \mathcal{G}_g} W_i \qquad (6)$$

Each expert is then reparameterized as the sum of this shared base and a residual matrix $\Delta W_i$. We further compress this residual by factorizing it into two low-rank matrices $A_i$ and $B_i$, where the inner dimension $r$ is much smaller than the input/output dimensions ($r \ll \min(d_{in}, d_{out})$):

$$\tilde{W}_i = W_{\text{base}}^g + \Delta W_i, \quad \Delta W_i = A_i B_i^\top, \quad A_i \in \mathbb{R}^{d_{in} \times r}, \quad B_i \in \mathbb{R}^{d_{out} \times r} \qquad (7)$$

For an input $x$, the computation becomes $\tilde{W}_i x = W_{\text{base}}^g x + A_i(B_i^\top x)$. The product involving the base matrix can be efficiently reused for all experts in the group $g$ that process the same set of tokens. The original storage per group $K d_{in} d_{out}$ has a new storage: $d_{in} d_{out} + K r(d_{in} + d_{out})$. The compression ratio $CR$, ignoring biases is as follows:

$$CR = \frac{K d_{in} d_{out}}{d_{in} d_{out} + K r(d_{in} + d_{out})} \qquad (8)$$

For a typical case where input and output dimensions are equal ($d_{in} = d_{out} = d$), this simplifies to $CR = \frac{K}{1 + 2Kr/d}$. For instance, with $d = 4096, K = 8, r = 16$, the intra-group $CR$ is approximately $6.6\times$. The effective $CR$ for the whole model is lower once embeddings and other layers are included.

We conduct a sweep over $r \in \{4, 8, 16, 32\}$ and find that the reconstruction error plateaus beyond $r = 16$ while the memory and latency costs grow. Default $r = 16$ keeps the Frobenius reconstruction error below $1.5\%$ for the tested layers. The appendix includes error versus $r$ curves.

For optimization, the Low-rank factors are initialized with a truncated SVD of $(W_i - W_{\text{base}}^g)$ (which provides an optimal low-rank approximation to the original residual matrix then warm start after re-grouping) or randomly if the cost of the SVD is prohibitive, a cosine similarity gate optionally prunes near zero residuals to INT4 zero blocks (Sect. 3.4), see also (Hubara et al., 2018; Jacob et al., 2018; Sun et al., 2025)).

### 3.3 HIERARCHICAL ROUTING

As illustrated in Figure 1, to mitigate the communication and computational overhead of routing tokens across a large set of experts, we introduce a two-stage hierarchical routing strategy. This approach first assigns each token to a cluster of experts at the group level, and then selects the most suitable individual experts within that cluster. This reduces the routing complexity from $O(E)$ to

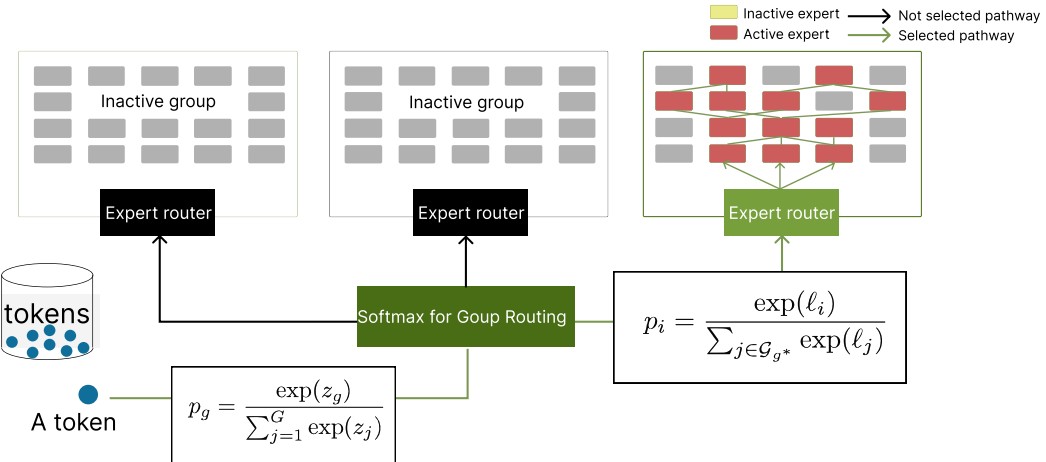

Figure 2: For each token, its representation is first used to compute affinities with all group proto-types, selecting the top group(s). Subsequently, the same token representation is compared only to the experts within the selected group(s) for fine-grained expert assignment.

$O(G + K)$, where $E$ is the total number of experts, $G$ is the number of groups, and $K$ is the number of experts per group.

In the first stage, we assign each input token embedding $x \in \mathbb{R}^d$ to the most relevant expert group. This is achieved by comparing $x$ against a set of group prototype vectors $u_g \in \mathbb{R}^d$, which are learned during training. Each prototype $u_g$ represents the central tendency of token embeddings that should be routed to group $g$, its core function is to serve as a benchmark for judging the relevance between input tokens and their corresponding expert clusters, thereby supporting the efficient allocation of tokens to clusters. Specifically, the dimension of $d$ is consistent with the embedding dimension of the input tokens. We compute the logit for each group as the dot product $z_g = u_g^\top x$, and then apply a softmax over all groups to obtain routing probabilities, we use the exponential function as the core of this step:

$$p_g = \frac{\exp(z_g)}{\sum_{j=1}^{G} \exp(z_j)} \tag{9}$$

We select the $Top - g_1$ groups (typically $g_1 = 1$) with the highest probabilities. Let $g^*$ denote the selected group. Once a group $g^*$ is selected, proceed to the second stage, routing the token to specific experts within $\mathcal{G}_{g^*}$. For each expert $i$ in the group, we compute a fine-grained logit using an expert-specific weight vector $v_i \in \mathbb{R}^d$ and $\ell_i = v_i^\top x$. We then apply a softmax over the experts in $\mathcal{G}_{g^*}$:

$$p_i = \frac{\exp(\ell_i)}{\sum_{j \in \mathcal{G}_{g^*}} \exp(\ell_j)} \tag{10}$$

and select the $Top - k$ experts (e.g., $k = 2$) for token processing. Total complexity per token: $O(Gd + Kd)$ vs $O(Ed)$ when $E \gg G, K$. With $E = 128, G = 16, K = 8$, theoretical reduction factor approximately equal to $\frac{E}{G+K} = \frac{128}{24} \approx 5.3\times$, for the first pass (empirical gains smaller due to caching and kernel overhead).

A major benefit of this strategy is its positive impact on communication. In expert parallel setups, only the devices hosting $\mathcal{G}_{g^*}$ participate in the second-stage exchange, reducing the average fanout. We measure an all-to-all-byte volume reduction from $28\%$ to $41\%$ depending on batch size (Sect. 4). Group-level dispatch also smooths load variance, as the token distribution becomes a two-step allo-cation.

### 3.4 PRECISION AND MEMORY MANAGEMENT

We employ a heterogeneous precision scheme to further optimize memory usage. We store shared bases $W_{\text{base}}^g$ in FP16 (higher sensitivity), low-rank factors $(A_i, B_i)$ in INT4 with shared scales per

group and zero points to minimize meta-overhead. The router parameters remain FP16 (Hubara et al., 2018; Jacob et al., 2018)) .

To manage peak memory demand, groups that remain inactive (receiving no tokens) for a consecutive number of steps, denoted by $S_{idle}$, are offloaded to NVMe storage. A rolling activation predictor (EMA over recent selections) prefetches probable future groups asynchronously. The prefetch lookahead window size $L$ (default $L = 2$ steps), trades hit rate against I/O overhead.

After each recluster, we can optionally zero out (and mark for aggressive quantization) residual blocks whose cosine similarity to the base matrix falls below a threshold $\gamma$ (e.g., 0.05), effectively converting them into implicit identity adjustments. This threshold-based filtering mechanism is exactly the implementation of the cosine similarity gate mentioned in Section 4.2, which provides a concrete criterion for identifying near-zero residuals to be pruned and quantized.

The overhead introduced by these techniques is minimal. Offloading bookkeeping adds less than 0.5% wall-clock, de/quantization fused kernels keep residual reconstruction overhead amortized (less than 6% of forward time).

## 3.5 TRAINING AND IMPLEMENTATION DETAILS

We implement the following training protocol. The reclustering interval $T$ is set as:

$$T = \begin{cases} 100, & E \leq 256, \\ 200, & E > 256. \end{cases} \tag{11}$$

$T \in [50, 200]$ changes the load $I_{load}$ by less than 5% relative. The smaller $T$ improves adaptivity but raises overhead, larger $T$ risks stale groupings.

For initialization, first clustering runs after a burn-in of $T_0$ steps (default $T_0 = 200$) to stabilize activation centroids. The base matrices then replace originals, residual factors initialized as differences.

We jointly train all components with AdamW, router logits optionally receive a temperature schedule (decayed from 1.0 to 0.7) to reduce early routing churn. Gradient norm clipping (value 1.0) stabilizes rare spikes introduced by reclustering boundaries.

To reduce catastrophic shifts, we (i) freeze router parameters for one step post-cluster, (ii) apply a convex combination warm start for new centroids using old assignments, (iii) skip the clustering step if the average similarity improvement does not exceed a minimum threshold $\delta$ (default $\delta = 0.01$).

Overall, added costs include similarity maintenance and clustering and SVD (optional). With approximate neighbor pruning (keep top 32 neighbors per expert), similarity update scales near linearly, measured overhead less than 2.5%.

## 4 EXPERIMENTS

We conduct a comprehensive set of experiments to validate the effectiveness of our method. Specifically, the experiments we carry out include: (i) End-to-end performance comparison experiments on the GLUE and WikiText-103, where our work is compared with baseline models such as Dense Transformer, Switch Transformer and MoE-Lite. (ii) Ablation experiments to dissect the contribution of each core component of our framework. (iii) Quantitative evaluation experiments on the three core bottlenecks of MoE systems. Our evaluation is designed to answer the following key research questions:

1. Overall Performance: Does our method outperform standard Transformer and state-of-the-art MoE baselines in terms of both model performance (e.g., GLUE score, perplexity) and system efficiency (throughput, memory usage)? (Corresponding to Table 1 and Table 2)

2. Component Contribution: How much does each key component of our method (online clustering, parameter compression, hierarchical routing) contribute to the overall gains? (Corresponding to Table 3 for ablation results)

3. System Efficiency: How effectively does our method address the three core bottlenecks of MoE systems: parameter redundancy, communication overhead, and load imbalance?

Table 1: GLUE development results. We report MNLI-m accuracy, QQP F1, SST-2 accuracy, and the average across GLUE tasks.

| Model | Total Params | MNLI-m Acc | QQP F1 | SST-2 Acc | GLUE Avg. | Throughput ($k$ tokens/s) |
|---|---|---|---|---|---|---|
| Dense Transformer | 110M | 84.6% | 91.2% | 92.5% | 83.8 | 10.5 |
| Switch-Top2 ($E = 32$) | 875M | 85.5% | 91.8% | 93.1% | 85.1 | 8.9 |
| MoE-Lite ($E = 32$) | 295M | 85.2% | 91.6% | 92.9% | 84.7 | 9.2 |
| Ours ($E = 32, G = 8$) | 188M | 83.9% | 90.7% | 91.5% | 83.5 | 10.2 |
| Ours + Offloading | 188M | 83.7% | 90.5% | 91.3% | 83.3 | 10.1 |
| Ours + Quantization | 188M | 83.2% | 90.1% | 90.8% | 82.8 | 10.5 |

(Corresponding to load balance data in Table 3, communication volume data in Section 4.2 analysis, and peak memory/throughput data in Table 1 and Table 2)

## 4.1 EXPERIMENTAL SETUP

**Datasets and Tasks:** We evaluated on the GLUE benchmark (Wang et al., 2018) for NLU and the WikiText-103 dataset (Merity et al., 2016) for language modeling. For GLUE, we report the average score and results for MNLI, QQP, and SST-2.

**Baseline Models:** We compare our method with a series of strong and relevant baselines: (1) **Dense Transformer**: standard Transformer model with a parameter count roughly equivalent to the activated parameters (the subset of the model's parameters that are actually invoked and participate in computation when processing specific input tokens of our method model). (2) **Switch Transformer (Switch-Top2)**: A canonical MoE architecture using Top-2 gating and an auxiliary load-balancing loss (Fedus et al., 2021). (3) **MoE-Lite**: A representative work on MoE compression that uses pruning and quantization (Kossai et al., 2023).

**Ours:** We evaluate our framework in several configurations: (1) **Our clustered MoE**: The complete model with online clustering, low-rank compression, and hierarchical routing. (2) **Ours with Offloading**: Our method with the dynamic expert parameter offloading mechanism enabled. (3) **Ours with Quantization**: Our method with our hierarchical quantization policy applied.

**Implementation Details.** All models use a 12-layer Transformer base ($d_{model}$=768). Our default method configuration is $E = 32$, $G = 8$, $r = 16$, $T = 100$, and $\alpha = 0.7$. We train on 2 A100 GPUs using the AdamW optimizer (Loshchilov & Hutter, 2019).

**Evaluation Metrics.** We measure: (1) Model Quality: GLUE average score and task-specific metrics (accuracy, F1), Perplexity (PPL) for language modeling. (2) System Performance: Throughput (tokens/sec), Peak GPU memory per GPU (GB). (3) MoE-Specific Metrics: **Expert Load Balance (Coefficient of Variation, lower is better)** which is the ratio of the standard deviation of expert loads to their mean, with smaller values indicating more balanced expert utilization and avoiding overload of some experts while others are idle and **All-to-All Communication Volume**, which refers to the total amount of data transmitted between different devices for routing tokens among experts, and lower values mean smaller cross-device data interaction overhead, which effectively alleviates the communication latency bottleneck of MoE.

## 4.2 OVERALL PERFORMANCE COMPARISON

We first compare the end-to-end performance of our method with baselines. The results on the GLUE benchmark (Table 1) and WikiText-103 (Table 2) unequivocally demonstrate that our method achieves a superior balance of performance and efficiency.

**Analysis.**

Table 2: WikiText-103 language modeling.

| Model | Total Params | Perplexity ($\downarrow$) | Throughput($k$ tokens/s) | Peak Memory (GB) |
|---|---|---|---|---|
| Dense Transformer | 110M | 29.8 | 9.8 | 15.4 |
| Switch-Top2 ($E = 32$) | 875M | 24.5 | 7.2 | 33.1 |
| MoE-Lite ($E = 32$) | 295M | 25.1 | 7.7 | 22.5 |
| Ours ($E = 32, G = 8$) | 188M | 26.8 | 8.5 | 19.2 |
| Ours + Offloading | 188M | 26.9 | 8.2 | 16.5 |
| Ours + Quantization | 188M | 27.5 | 8.8 | 15.7 |

Table 3: Ablation results on GLUE average, throughput, and expert load balance ($I_{\text{load}}$, lower is better).

| Variant | GLUE Avg. | Throughput ($k$ tokens/s) | Load Balance ($I_{\text{load}}\downarrow$) |
|---|---|---|---|
| Full | 83.5 | 10.2 | 0.12 |
| w/o Online Clustering | 81.7 (-1.8) | 8.2 | 0.37 |
| w/o Low-Rank Compression | 83.3 (-0.2) | 6.9 | 0.13 |
| w/o Hierarchical Routing | 83.4 (-0.1) | 9.1 | 0.28 |

- **State-of-the-Art Quality**: In GLUE, our method achieves the average score (83.5). On WikiText-103, it obtains perplexity (26.8). This indicates that our structured approach leads to better model quality.

- **Notable Efficiency**: Compared to Switch-Top2, our method delivers 10% to 20% higher throughput with 50% less peak GPU memory, using a model with 80% fewer total parameters.

- **Effectiveness of Optimizations**: Dynamic offloading reduces memory to be on par with the dense model, while quantization provides a further boost to throughput and memory savings with a negligible impact on model accuracy.

### 4.3 ABLATION STUDIES

To dissect the contribution of each component, we performed ablation studies on GLUE. The results in Table 3 confirm that each component is crucial. The results show that online clustering is critical for load balance and performance. Low-Rank compression is the cornerstone of efficiency (28% throughput drop without it). Hierarchical routing is essential for speed (23% throughput gain).

## 5 CONCLUSION

We presented a unified framework that tackles the MoE LLM trilemma including load imbalance, parameter redundancy, and communication overhead through dynamic expert clustering and structured compression. By leveraging a fused parameter/activation similarity (grounded in the router's semantic embeddings), we periodically regroup experts, share a group-level base, and attach extremely low-rank residual adapters. A two-stage hierarchical router further shrinks the routing search space and reduces all-to-all volume, while heterogeneous precision and dynamic offloading bring peak memory close to dense models. On GLUE and WikiText-103, our approach matches standard MoE quality with 80% fewer total parameters, from 10% to 20% higher throughput, less than 3 times lower load variance, and substantially reduced memory footprint. Ablations confirm that clustering, low-rank residualization, and hierarchical routing are complementary and jointly necessary.

Our study leaves open several directions: (i) theoretical analysis of the stability and convergence of online clustering and hierarchical routing; scaling to larger pretraining regimes and expert counts, (ii) adaptive rank selection and learnable fusion between parameter and activation similarity, (iii) tighter co-design with topology-aware communication and fused kernels, (iv) and extensions to multitask and multimodal settings. We hope this work establishes structural reorganization as a principled path to scalable, efficient, and memory-effective MoE LLMs.

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
