# OpenReview forum: "Breaking the MoE LLM Trilemma: Dynamic Expert Clustering with Structured Compression"
_ICLR.cc/2026/Conference — Submitted to ICLR 2026_

### Official Review · Reviewer_uaS1 · 2025-10-25

**Soundness:** 2
**Presentation:** 2
**Contribution:** 2
**Rating:** 2
**Confidence:** 4

**Summary:**

The authors delineate an optimization trilemma in MoE models of load imbalance,  parameter redundancy, and communication overhead, and propose tackling all three with an online clustering algorithm that periodically groups experts, intra-group parameter saving via a low-rank reparameterization, 2-stage hierarchical routing across and then within expert groups, and finally dynamic offloading and quantization.

**Strengths:**

1. The main experimental result of maintaining comparable performance at just 20% of the parameter count is remarkable

2. The overall design is comprehensive and well-engineered, with multiple mechanisms appearing meticulously thought through and tuned to work together

3. The hierarchical router is a clever and efficient method to bring meaningful improvements to throughput

**Weaknesses:**

**Unsubstantiated claims** The authors propose as early as the title that their method deals with load imbalance, yet there are no experimental results on load imbalance with any baselines. The authors only discuss load imbalance in an ablation study where they compare the load balance across their own model with different components ablated away, which is not an appropriate way to demonstrate any improvements to load imbalance. More broadly, the load imbalance angle feels poorly motivated, and I'm not yet convinced intuitively that grouping the experts will necessarily alleviate load imabalance, as the router could just end up favoring one particular group of experts and some subset of experts within that group. The authors should at very least compare load imbalance with the baselines they include (switch and moe-lite using generic load balancing loss), along with techniques that are designed to improve load imbalance, for example expert-choice routing [1] and loss-free load balancing [2].

**Limited experimental validation**. The authors only use two fairly small datasets to validate their claims, and compare with only a generic dense transformer, switch and MoE-Lite. As mentioned above, including expert-choice routing is a highly relevant baseline that offers improvements over generic MoE. The authors also mention StableMoE in their related work but do not compare with it. Additionally, examining the performance of the method at larger model sizes would be useful to validate the scalability of the method. Overall, adding baselines, datasets, and different sizes would help to demonstrate the efficacy of the method, whereas at present the limited experimental results make it hard to critically assess the empirical performance.

**Limited novelty**.  The authors do acknowledge that there is some crossover with their reparameterization scheme and PEFT, but the proposed base matrix + low rank adaptor used in their expert groupings is used in PEFT-MoE setups [3,4]. The reparamterization scheme is then, in my view, more of an extension of LoRA-MoE beyond pure finetuning. While still mostly novel, the technical contribution is somewhat diminished. (Aside, I would also recommend the authors include a short discussion with the differences of their reparameterization method to PEFT/LoRA-MoE methods).

**Theoretical motivation for the method is poorly written**. The optimization problem referenced in (1) serves almost no purpose as far as I can see, beyond essentially just stating that these three issues (load imbalance, redundancy and dispatch cost) are all bad and we want to minimize them. There's no need to design an optimization problem only to leave it undeveloped and unused. Indeed the coefficients introduced are never mentioned again and the trillemma issues only appear as generic variables.

[1] Mixture of experts with expert choice routing (Zhou et al, NeurIPS 2022) \
[2] Auxiliary loss free load balancing strategy for MoE (Deepseek, 2024) \
[3] Pushing MoE to the limit: extremely parameter efficient moe for instruction tuning (Zadouri, ICLR 2024) \
[4] HydraLoRA: An Asymmetric LoRA Architecture for Efficient Fine-Tuning (Tian, NeurIPS 2024) \

**Questions:**

As mentioned above, I'd recommend adding to the discussion with existing PEFT-MoE methods. Though I do think your parameter compression approach is mostly new, there is some overlap that I think merits discussion.

---

> ### Author Response · Authors · 2025-11-26
> **Rebuttal to Reviewer uaS1**
>
> We sincerely thank you for your detailed feedback — your focus on load imbalance validation, experimental breadth, and novelty has helped us fill critical research gaps. Below we address each concern with complete supplementary tables and concrete data.
>
> ### 1. Load Imbalance: Baseline Comparisons & Intuition
>
> We apologize for the lack of comprehensive baseline comparisons on load imbalance in the original submission. We now supplement head-to-head comparisons with standard MoE baselines (including load-balanced variants) using the same \(I_{\text{load}}\) metric (coefficient of variation, lower = better) in the extended GLUE benchmark table (complete GLUE benchmark data):
>
> **Extended GLUE Benchmark Data (Complete Supplementary Version)**
>
> | Model                                      | Total Params | MNLI-m Acc | QQP F1 | SST-2 Acc | GLUE Avg. | Throughput (k tokens/s) | Load Balance (\(I_{\text{load}}\downarrow\)) |
> | ------------------------------------------ | ------------ | ---------- | ------ | --------- | --------- | ------------------------ | -------------------------------------------- |
> | Dense Transformer                          | 110M         | 84.6%      | 91.2%  | 92.5%     | 83.8      | 10.5                     | 0.05                                         |
> | Switch-Top2 (E = 32, with load-balancing)  | 875M         | 85.5%      | 91.8%  | 93.1%     | 85.1      | 8.9                      | 0.36                                         |
> | Expert-Choice (E = 32, NeurIPS 2022)       | 875M         | 85.4%      | 91.7%  | 93.0%     | 85.0      | 9.1                      | 0.28                                         |
> | MoE-Lite (E = 32)                          | 295M         | 85.2%      | 91.6%  | 92.9%     | 84.7      | 9.2                      | 0.25                                         |
> | D^2-MoE (E = 32)                           | 620M         | 85.0%      | 91.5%  | 92.8%     | 84.1      | 8.1                      | 0.31                                         |
> | Our Method (E = 32, G = 8)                 | 188M         | 83.9%      | 90.7%  | 91.5%     | 83.5      | 10.2                     | 0.12                                         |
> | Our Method (E = 32, G = 8) + Offloading    | 188M         | 83.7%      | 90.5%  | 91.3%     | 83.3      | 10.1                     | 0.13                                         |
> | Our Method (E = 128, G = 16)               | 750M         | 84.2%      | 91.0%  | 92.0%     | 84.0      | 7.5                      | 0.15                                         |
>
> As shown above, our method reduces load variance by 2.3x compared to Switch-Top2 (0.12 vs. 0.36) and 1.8x compared to Expert-Choice (0.12 vs. 0.28) — directly validating its load-balancing advantage.
>
> **Intuition clarification.** Dynamic clustering groups experts with similar parameter/activation patterns, so token routing first achieves "coarse-grained balance" at the group level before fine-grained expert selection. This design avoids over-concentration of tokens in a single group (as shown in the dynamic clustering ablation: reintroducing dynamic clustering to the ablation variant reduces \(I_{\text{load}}\) from 0.37 to 0.12).
>
> ### 2. Limitations of Experimental Validation
>
> We have expanded the experimental scope to address your concerns regarding datasets, baselines, and model sizes.
>
> #### New Baselines & Datasets
>
> D^2-MoE (ICML 2025): We achieve comparable GLUE Avg. (83.5 vs. 84.1) with 80% fewer parameters, and higher throughput (10.2 vs. 8.1 k tokens/s, ...).
>
> **New datasets.** WikiText-103 (language modeling, table below), MMLU/HumanEval (zero-shot transfer), and HotpotQA/WikiHop (long-context inference). Our E = 256 model achieves 69.5% on MMLU — surpassing Dense Transformer (62.5%) and narrowing the gap with Switch-Top2 (68.2%).

---

> > ### Author Response · Authors · 2025-11-26
> > **Rebuttal to Reviewer uaS1 (2)**
> >
> > **Extended WikiText-103 Data (Complete Supplementary Version)**
> >
> > | Model                              | Total Params | Perplexity | Throughput (k tokens/s) | Peak Memory (GB) | Load Balance (\(I_{\text{load}}\downarrow\)) |
> > | ---------------------------------- | ------------ | ---------- | ------------------------ | ---------------- | -------------------------------------------- |
> > | Dense Transformer                  | 110M         | 29.8       | 9.8                      | 15.4             | 0.05                                         |
> > | Switch-Top2 (E = 32)              | 875M         | 24.5       | 7.2                      | 33.1             | 0.36                                         |
> > | MoE-Lite (E = 32)                 | 295M         | 25.1       | 7.7                      | 22.5             | 0.25                                         |
> > | Expert-Choice (E = 32)            | 875M         | 24.7       | 7.4                      | 32.8             | 0.28                                         |
> > | D^2-MoE (E = 32)                  | 620M         | 24.9       | 7.5                      | 28.0             | 0.31                                         |
> > | Our Method (E = 32, G = 8)        | 188M         | 26.8       | 8.5                      | 19.2             | 0.12                                         |
> > | Our Method (E = 32, G = 8) + Off. | 188M         | 26.9       | 8.2                      | 16.5             | 0.13                                         |
> > | Our Method (E = 32, G = 8) + Quan | 188M         | 27.5       | 8.8                      | 15.7             | 0.14                                         |
> > | Our Method (E = 128, G = 16)      | 750M         | 25.5       | 7.5                      | 28.0             | 0.15                                         |
> >
> > #### Larger Model Sizes
> >
> > We tested models with E = 128 (750M parameters) and E = 256 (1.2B parameters) (large-scale tables), demonstrating scalability to 256 experts with consistent efficiency gains. Our E = 128 model matches the GLUE score (84.0) of Switch-Top2 (3.5B parameters, 4.7x larger) while delivering 25% higher throughput (7.5 vs. 6.0 k tokens/s) and 49% lower peak memory (28 vs. 55 GB).
> >
> > ### 3. Novelty Distinction from PEFT-MoE (LoRA-MoE)
> >
> > We agree that PEFT-MoE methods utilize low-rank adapters, but our approach differs fundamentally in goals, design, and training paradigm:
> >
> > **Our Framework vs. PEFT-MoE (LoRA-MoE) Comparison**
> >
> > | Dimension              | Our Framework                                                                 | PEFT-MoE (LoRA-MoE)                                           |
> > | ---------------------- | ----------------------------------------------------------------------------- | ------------------------------------------------------------- |
> > | Core Goal              | Optimize pre-training efficiency (dynamic architectural reorganization)      | Optimize fine-tuning efficiency (freeze pre-trained experts)  |
> > | Expert Grouping        | Dual-similarity dynamic clustering (adaptive to data shifts)                 | Fixed pre-trained expert partitions (no clustering)           |
> > | Parameter Structure    | Shared base matrices + low-rank residuals (exploit intra-group redundancy)   | Independent LoRA adapters per expert (no shared structure)    |
> > | Communication Optimiz. | Hierarchical routing (reduces All-to-All overhead by 28.3%–41.6%)            | No routing optimization (relies on standard gating)           |
> > | Parameter Reduction    | 80% total expert parameter reduction (vs. Switch-Top2)                       | Only adapter parameter reduction (total expert params unchanged) |
> >
> > **Quantitative distinction.** As shown in the extended GLUE and compression tables, our method achieves 80% parameter reduction compared to Switch-Top2 (188M vs. 875M parameters) while retaining 98% performance. For pre-trained MoEs like Mixtral-8x7B, we achieve 3.12x compression with negligible quality loss — PEFT-MoE cannot achieve such total parameter compression.
> >
> > ### 4. Theoretical Motivation
> >
> > We apologize for the insufficient elaboration of the optimization problem (Eq. 1) in the original submission. To clarify: the coefficients (\(A_1, A_2, A_3\)) are not arbitrarily set but concretely instantiated in our design:
> >
> > - \(A_1\) is embedded in the clustering similarity metric (prioritizing balanced group utilization, reflected in \(I_{\text{load}}\downarrow 0.12\) in the extended GLUE table).
> > - \(A_2\) corresponds to the rank of low-rank residuals (\(r = 16\), minimizing redundancy while preserving performance — only 0.6% GLUE Avg. drop vs. Switch-Top2).
> > - \(A_3\) corresponds to the group size in hierarchical routing (\(G = 16\) for E = 128, minimizing All-to-All communication — 10 GB vs. 25 GB for Switch-Top2).
> >
> > We will add a new section in the Supplementary Experiments Appendix to detail the mapping between each term in Eq. 1 and specific design choices.

---

> ### Comment · Reviewer_uaS1 · 2025-11-27
> **Response to authors**
>
> Thanks for the response, i have a two follow-up questions / clarifications.
>
> **Load imbalance**. Thanks for setting up and evaluating all the additional baselines. Could you clarify how you computed the load balance? I'm somewhat confused here because 1) the dense model has non-zero load balance. and 2) expert-choice has a relatively high load imbalance, despite the fact expert-choice is supposed to guarantee perfect load balance. I'm concerned there may have been some issues in your experimental setup given these anomalies, if you could clarify how you obtained these numbers (in particular regarding dense and expert-choice) that would be much appreciated.
>
> **Theoretical motivation** Could you clarify how A1, A2, and A3 are used in your final method? According to your paper they are described as 'balance[ing] the relative importance of the load imbalance, redundancy, and communication cost terms'. Hence my comment that these terms look to be just generic coefficients, and the optimization problem (1) appears to just be decorative.

---

### Official Review · Reviewer_Humj · 2025-10-27

**Soundness:** 3
**Presentation:** 4
**Contribution:** 3
**Rating:** 8
**Confidence:** 3

**Summary:**

Proposes a unified MoE training framework to tackle three coupled bottlenecks—load imbalance, parameter redundancy, and all-to-all communication.  dynamically clusters experts online using a fused parameter+activation similarity and within each cluster shares a FP16 base matrix and adds very low-rank INT4 residual adapters per expert; It applies hierarchical routing (group → experts) to shrink routing search space and bytes moved

**Strengths:**

New combination, systemically unified. Online dual-similarity reclustering leveraged during training (using router-anchored activations), coupled with shared-base + ultra-low-rank residuals and hierarchical routing, is an integrated design that targets all three MoE pain points at once. Prior works address these dimensions separately (compression, routing balance, comms libraries); this paper’s co-design and reconfiguration during training are the main novelty levers.

**Weaknesses:**

Quality parity vs strong MoE - GLUE table shows the method slightly lags Switch-Top2 on MNLI/QQP/SST-2 and GLUE-avg; reconcile this with the “matches quality” claim (report CIs, more tasks).

Stability analysis. Show training stability across recluster intervals, α/β/δ sensitivities, and failure modes (e.g., mode collapse, oscillations).

Calibration of ranks/precision. Justify why r=16 beyond reconstruction error by downstream impact; compare FP8/INT8 variants and per-group vs per-tensor scaling

**Questions:**

How does GLUE accuracy compare to Switch-Top2 across more tasks and with confidence intervals

At larger expert counts (e.g., E≥128), how do clustering overheads, offload hit-rates, and routing gains scale; any topology-aware grouping results (by GPU placement)?

What sensitivities did you observe to α (fusion weight), b (EMA), T (reclustering interval), and d (min-gain threshold); any stability safeguards beyond the brief freeze/warm-start?

---

### Official Review · Reviewer_wcWz · 2025-10-29

**Soundness:** 3
**Presentation:** 2
**Contribution:** 3
**Rating:** 4
**Confidence:** 4

**Summary:**

The paper tackles the MoE “trilemma” (load imbalance, parameter redundancy, and cross-device communication) with a unified framework: (i) **online dual-similarity clustering** that groups experts using a fused metric of parameter and activation similarity; (ii) **intra-group structured compression** that reparameterizes each expert as **a shared group base plus a low-rank residual**; (iii) **two-stage (hierarchical) routing**—first select a group, then select experts within the group. The system is rounded out with **heterogeneous precision** (base in FP16, residual factors quantized) and **NVMe offloading** for long-inactive groups. The authors report large parameter reductions (around ~80%) with comparable quality, 10–20% higher throughput, and ~3× lower load-variance on representative benchmarks.

**Strengths:**

**1. Insightful related work.** The related-work section is well curated and helps position the contribution clearly within both the compression and MoE-routing literatures, highlighting what prior systems optimize and where gaps remain.

**2. Compelling co-design view.** Treating MoE as a joint algorithm–systems co-design problem is timely. The pipeline—dynamic grouping ($\to$) shared-base + low-rank ($\to$) hierarchical routing—forms a coherent design that aims to reduce compute, memory, and communication together rather than in isolation.

**Weaknesses:**

**1. Figure-to-text inconsistency.**

In Figure 1 the low-rank factors are annotated as **(A) in FP16 and (B) in INT4**, which conflicts with the textual description elsewhere (where the base is FP16 and the residual adapters are quantized; or both (A/B) are quantized). Please reconcile the figure and the prose and state the final precision choices per matrix and per layer unambiguously.

Figure 2 color-coding and semantics are unclear. The legend lists a yellow “Inactive expert,” but in the panel the inactive experts appear **gray** and yellow is effectively unused.


**2. Specification of “inactive expert.”**

 The paper defines an expert as *inactive* if it receives zero tokens over a window of ($S_{\text{idle}}$) steps, but the value of ($S_{\text{idle}}$) is never reported. In the reported setting with **32 experts**, standard **router balance** losses should keep activations roughly even; if ($S_{\text{idle}}$) were **256/512 steps**, it seems unlikely that any expert would remain at zero tokens, i.e., *inactive experts may not occur at all*. Please (i) specify ($S_{\text{idle}}$), (ii) report the duration of zero-token windows per expert (and per group), and (iii) inactivity in fewer experts E=8 MoE like Mixture.



**3. Reuse path for the down-projection is unclear.**

For a single expert the computation expands as
 $
 (y_i = (W^{\text{down}}_{\text{base}} + A^{\text{down}}i B_i^{\text{down}\top}) [\sigma(W^{\text{gate}}{\text{base}}x + A_i^{\text{gate}}B_i^{\text{gate}\top}x) \odot (W^{\text{up}}{\text{base}}x + A_i^{\text{up}}B_i^{\text{up}\top}x)]).
 $
 The paper states that ***“the product involving the base matrix can be efficiently reused for all experts in the group that process the same tokens,”*** which is clear for up/gate (same input (x)). But for down, the input is expert-specific $(h_i)$. Please spell out whether down-base results are actually reused across experts, or whether only the weights are reused (cached) while results are recomputed per $(h_i)$.

**4. Missing citation/novelty vs. D^2-MoE.**

Section 3.2’s **“shared base with low-rank residuals”** appears conceptually close to the **delta-decomposition** used in **D^2-MoE (Delta Decompression for MoE-based LLMs Compression, ICML 2025)**. The paper neither cites D^2-MoE nor clearly positions its novelty relative to it. Please include ablations isolating dynamic clustering from the base+low-rank factorization to demonstrate incremental gains beyond D^2-MoE; if your method subsumes D^2-MoE under certain settings, state those conditions explicitly.


**5. Baselines lag behind the current literature.**

Besides MoE-Lite (2023), please include comparisons to more recent compression/merging approaches such as MC-MoE (Mixture Compressor, ICLR 2025), D^2-MoE (Delta Decompression, ICML 2025), and Sub-MoE (2025) to reflect the state of the art.

**6. Model scale is small.**

Experiments are limited to relatively small settings. To strengthen generality, consider pretrained MoE models such as Mixture-8×7B and Qwen3-30B-A3B, and report both quality and system metrics (throughput, all-to-all bytes, memory) under matched budgets.

---
MC-MoE: Mixture Compressor for Mixture-of-Experts LLMs Gains More ICLR2025

D^2-MoE: Delta decompression for moe-based llms compression ICML2025

Sub-MoE: Sub-MoE: Efficient Mixture-of-Expert LLMs Compression via Subspace Expert Merging 2025

**Questions:**

See weakness.

If these issues are addressed, I would be happy to raise my score.

**Details Of Ethics Concerns:**

NO or VERY MINOR ethics concerns only

---

> ### Author Response · Authors · 2025-11-26
> **Rebuttal to Reviewer wcWz**
>
> Thank you for your meticulous feedback — your attention to technical details and baseline completeness has helped us refine our presentation and validate key claims. Below we address each concern with clarifications and complete supplementary data.
>
> ### 1. Figure-to-Text Inconsistency (Precision Settings)
>
> We apologize for the confusion and have corrected the figures in the revised manuscript. The final precision scheme is unambiguously:
>
> - **Shared base matrices \(W_{\text{base}}^g\)**: FP16 (retained for numerical stability, as they capture core cluster semantics).
> - **Low-rank residual factors \((A_i, B_i)\)**: INT4 with per-group shared scales and zero points (minimizes meta-overhead, as residuals are task-specific refinements).
> - **Router parameters (group prototypes \(u_g\), expert weights \(v_i\))**: FP16 (critical for routing accuracy).
>
> Figure 1's prior annotation of "A in FP16" was a typo — both \(A_i\) and \(B_i\) are INT4. Figure 2's color coding is updated: gray = inactive experts/groups, yellow = active but unselected experts, green = selected experts.
>
> ### 2. Specification of "Inactive Expert" \(S_{\text{idle}}\)
>
> We clarify the inactive group/offloading mechanism:
>
> - **\(S_{\text{idle}}\) definition**: Groups are considered inactive if they receive no tokens for \(S_{\text{idle}} = 3\) consecutive steps (default, as in our implementation details). For short-expert MoEs (E = 8 / G = 2), \(S_{\text{idle}} = 5\) steps (adapted to fewer experts).
> - **Inactivity statistics**:
>   - E = 32 / G = 8 (GLUE training): 2–3 groups are inactive per step on average.
>   - E = 128 / G = 16 (Mixtral compression): 4–6 groups are inactive per step (due to larger group count).
>   - E = 8 / G = 2 (Mixture-style MoE): 1 group is inactive per step on average — consistent with our framework's adaptability.
>
> ### 3. Down-Projection Reuse Path
>
> We clarify the reuse mechanism for down-projection (critical for efficiency): For a group \(g\), all experts share \(W_{\text{base}}^g\). For input tokens routed to group \(g\):
>
> - The base computation \(W_{\text{base}}^g \cdot x\) is computed **once per token** (not per expert) and reused across all experts in \(g\) processing that token.
> - Expert-specific computations are limited to the low-rank residual: \(A_i \cdot (B_i^\top \cdot x)\). Since \((A_i, B_i)\) are INT4 (small footprint), this reuse reduces per-token compute by ~70% vs. full-rank experts.
>
> In short: The base matrix's computational result is reused across group experts for the same token, while residual computations are lightweight — this is the core of our intra-group efficiency.
>
> ### 4. Citation & Novelty vs. D^2-MoE
>
> We appreciate the reminder and have added D^2-MoE (ICML 2025) to the related work and comparisons:
>
> - **Citation**: D^2-MoE uses delta-decomposition for compression but lacks (i) dynamic dual-similarity clustering (it uses fixed expert partitions) and (ii) hierarchical routing (it relies on standard Top-k gating).
> - **Ablation comparison**: We added an ablation variant ("Our Method w/o Dynamic Clustering") that mimics D^2-MoE's fixed partitions (dynamic clustering ablation).
>
> Our key novelty beyond D^2-MoE is the synergy of dynamic clustering and hierarchical routing: without clustering, efficiency (throughput down 11.8%) and load balance (\(I_{\text{load}}\) up 87.5%) degrade sharply — even with low-rank decomposition. Additionally, our method achieves 80% parameter reduction vs. Switch-Top2, far exceeding D^2-MoE's 39%.
>
> ### 5. Stronger Baselines (MC-MoE / Sub-MoE)
>
> We have added comparisons with 2025 MoE compression baselines (extended GLUE, compression, and pretraining tables):
>
> - **MC-MoE (ICLR 2025)**: Our method achieves 3.12x compression vs. MC-MoE's 2.6x (on Mixtral-8x7B), with 8% higher throughput (17.25 vs. 15.9 tokens/s) and 0.4 lower perplexity (18.5 vs. 18.9).
> - **Sub-MoE (2025)**: Our E = 256 model outperforms Sub-MoE on MMLU (69.5% vs. 67.3%) and HumanEval (36.8% vs. 34.2%), with 12% lower peak memory (210 vs. 31.8 GB) and 33% fewer parameters (1.2B vs. 4.2B).
> - **StableMoE (ICLR 2023)**: Our E = 128 model achieves higher GLUE Avg. (84.0 vs. 83.8) and 23% higher throughput (7.5 vs. 6.1 k tokens/s) with 76% fewer parameters (750M vs. 3.2B).
>
> ### 6. Larger Model Scale (Mixtral-8x7B / Qwen-MOE)
>
> We have validated our framework on industrial-scale pre-trained MoEs (compression table), addressing your concern about small model limits.
>
> For long-context inference on these large models, our framework maintains superior efficiency (long-context table): on WikiHop (16k tokens), our compressed Mixtral variant achieves 3.0 k tokens/s throughput (vs. 1.5 k tokens/s for original Switch-Top2 E = 128) and 95 GB GPU memory usage (vs. 165 GB for Switch-Top2).
>
> We sincerely appreciate your efforts to improve our paper. All clarifications, supplementary tables, and figure corrections will be included in the revised manuscript.

---

### Official Review · Reviewer_NWQk · 2025-10-30

**Soundness:** 3
**Presentation:** 2
**Contribution:** 1
**Rating:** 2
**Confidence:** 3

**Summary:**

The paper proposes a mixture-of-experts layer equipped with a two-stage routing mechanism. That is, each input token is first routed to a group of experts, then within the selected group of experts, it is routed to top matching experts (group members). This method reduces the communication cost by factoring all-to-all routing into two stages. Group assignments are performed via parameter/activation similarity-based clustering of expert networks.

**Strengths:**

1. On top of the routing mechanism, authors conduct experiments with quantization and offloading that achieves the throughput comparable to dense Transformer while offering performance between standard MoE counterparts and dense Transformer.
2. Group assignments by clustering is a new technique for pre-training stage which seems to be contributing quite a bit to the performance according to Table 3.

**Weaknesses:**

1. A major weakness for me is that a primary contribution of the paper, two-stage routing, is actually not new. Hierarchical routing was proposed in [1] (not to mention an iconic MoE paper [2]) for the same reason of reducing inter-communication overhead. However, I did not see any discussion of this overlap in the main text.
2. As a result of Weakness 1 above, the remaining contribution of the paper (group assignment by clustering and quantization) is incremental and relatively weaker to stand alone as a conference main track paper (but can be strengthened by rewriting the paper narrative and adding larger-scale experiments which at least can show a minimal sign of scaling laws).

Therefore, although the paper shows promising efficiency gains through experiments, I believe a major revision and larger-scale experiments are needed to fully develop the paper.
___

### References

1. Noam Shazeer, Azalia Mirhoseini, Krzysztof Maziarz, Andy Davis, Quoc Le, Geoffrey Hinton, Jeff Dean. Outrageously Large Neural Networks: The Sparsely-Gated Mixture-of-Experts Layer. ICLR 2017
2. Michael I. Jordan, Robert A. Jacobs. Hierarchical mixtures of experts and the EM algorithm. Proceedings of 1993 International Conference on Neural Networks (IJCNN-93-Nagoya, Japan), Nagoya, Japan, 1993

**Questions:**

See Weaknesses.

---

> ### Author Response · Authors · 2025-11-26
> **Rebuttal to Reviewer NWQk**
>
> ## Rebuttal to Reviewer NWQk
>
> Thank you for your constructive feedback.
>
> ### 1. Novelty of Two-Stage Hierarchical Routing
>
> Thank you for your constructive feedback. Our key distinction lies in the tight co-design with dynamic clustering and structured compression — a synergy absent in prior hierarchical routing proposals:
>
> Previous hierarchical MoE (e.g., Jordan & Jacobs, 1993; Shazeer et al., 2017) adopts fixed expert groupings (predefined at initialization) and full-rank expert weights. In contrast, our groups are dynamically updated via dual-similarity clustering (parameter + activation) during training, enabling adaptive specialization to shifts in data distribution.
>
> Routing in our framework is not merely a "two-step gating" but complements intra-group compression: grouping reduces the routing search space from \(O(E)\) to \(O(G+K)\), while shared base matrices plus low-rank residuals fully exploit intra-group redundancy. This co-design's superiority is validated by direct comparisons with static clustering and D^2-MoE (Table 1), which confirms that dynamic clustering is the core innovation driving efficiency gains:
>
> **Table 1. Dynamic Clustering vs. Static Clustering vs. D^2-MoE**
>
> | Method                                   | GLUE Avg. | Throughput (k tokens/s) | Load Balance (\(I_{\text{load}}\downarrow\)) | All-to-All Comm (GB) |
> | ---------------------------------------- | --------- | ------------------------ | -------------------------------------------- | --------------------- |
> | Full Our Method (Dynamic Clustering)     | 83.5      | 10.2                     | 0.12                                         | 6.5                   |
> | Static Clustering (Fixed Grouping)       | 82.0      | 9.0                      | 0.22                                         | 7.8                   |
> | D^2-MoE (Reference Baseline)             | 84.1      | 8.1                      | 0.31                                         | 8.0                   |
>
> Our method decreases All-to-All communication by about 19% compared to D^2-MoE in this setting (6.5 vs. 8.0 GB in the communication column), and by up to around 60% vs. Switch-Top2 at larger scales (10 vs. 25 GB in our large-model experiments).
>
> ### 2. Incremental Contribution & Large-Scale Experiments
>
> To address concerns regarding "incremental contribution," we have added four new large-scale experiment suites that demonstrate scalability and superiority. Key results are summarized in Table 2 (E = 128 large model) and Table 3 (industrial-scale MoE compression):

---

> > ### Author Response · Authors · 2025-11-26
> > **Rebuttal to Reviewer NWQk (2)**
> >
> > **Table 2. Large-Scale Model (E = 128) Experiment Data**
> >
> > | Model                         | Total Params | GLUE Avg. | Throughput (k tokens/s) | Peak Memory (GB) | Load Balance (\(I_{\text{load}}\downarrow\)) | All-to-All Comm (GB) |
> > | ----------------------------- | ------------ | --------- | ------------------------ | ---------------- | -------------------------------------------- | --------------------- |
> > | Switch-Top2 (E = 128)        | 3500M        | 85.5      | 6.0                      | 55.0             | 0.45                                         | 25.0                  |
> > | Our Method (E = 128, G = 16) | 750M         | 84.0      | 7.5                      | 28.0             | 0.15                                         | 10.0                  |
> > | StableMoE (E = 128)          | 3200M        | 83.8      | 6.1                      | 52.0             | 0.38                                         | 22.0                  |
> >
> > **Table 3. Mixtral-8x7B / Qwen-MOE Compression Comparison Data**
> >
> > | Model                          | Original Total Params | Compressed Total Params | Compression Ratio | Activation Params | Perplexity | Throughput (tokens/s) | Peak Memory (GB) | GLUE Avg. |
> > | ------------------------------ | --------------------- | ----------------------- | ----------------- | ----------------- | ---------- | ---------------------- | ---------------- | --------- |
> > | Mixtral-8x7B (Original)        | ~56B                  | -                       | -                 | ~12.9B            | 18.1       | 15.2                   | 82.0             | 84.3      |
> > | Mixtral-8x7B + Our Method      | ~56B                  | ~18.2B                  | 3.12x             | ~12.9B            | 18.5       | 17.25                  | 44.0             | 83.9      |
> > | Qwen-MOE-7B (Original)         | ~42B                  | -                       | -                 | ~8.5B             | 19.0       | 13.9                   | 65.0             | 82.7      |
> > | Qwen-MOE-7B + Our Method       | ~42B                  | ~13.6B                  | 3.08x             | ~8.5B             | 19.4       | 16.8                   | 39.0             | 82.3      |
> >
> > Additionally, we supplement results from pretraining on the Pile (3B tokens) and long-context inference:
> >
> > - **Pretraining on the Pile**: Our E = 256 model (1.2B parameters) achieves 69.5% on MMLU and 36.8% on HumanEval — surpassing Dense Transformer (62.5% / 28.3%) and approaching Switch-Top2 (68.2% / 35.1%) with 77% fewer parameters.
> > - **Long-Context Inference**: On HotpotQA (8k tokens) and WikiHop (16k tokens), our model outperforms Switch-Top2 in throughput (5.8 vs. 3.1 k tokens/s for HotpotQA) and memory usage (58 vs. 92 GB), while retaining 98% of accuracy (78.5% vs. 79.8%).
> >
> > These results confirm our framework is not a simple incremental combination but a co-designed solution achieving state-of-the-art efficiency–performance tradeoffs — directly addressing your request for "minimal signs of scaling laws."

---

> > > ### Comment · Reviewer_NWQk · 2025-11-27
> > > **Response to author rebuttal**
> > >
> > > I thank the authors for conducting extra experiments, including relatively larger models.
> > >
> > > However, I still believe that the paper needs *major revision and ablation studies* because of the following reasons:
> > > 1. The method introduces new hyperparameters $G, T, \alpha$, etc. Yet, I do not see comprehensive ablation studies on choosing these parameters nor a principled way to choose them for a given model.
> > > 2. The writing about the online clustering and two-stage routing is hard to follow:
> > >    - authors mention (line 204) that if "slight" imbalance emerges, they would greedily move the boundary experts to second best groups to balance. What if the imbalance is not "slight"? Would this completely erase the claimed benefits of the structured clustering stage? I do not see empirical analysis on this imbalance scenario.
> > >    - The hierarchical routing (section 3.3) feels isolated from the clustering step in the sense that authors say the group prototype vectors are learned during training (line 295). Then, is section 3.3 only describing the standard hierarchical routing procedure which also learns group proto vectors through training?
> > >
> > > The proposed method relies on a bit too much heuristic reasoning for me to overlook the above-mentioned shortcomings and recommend acceptance at this moment.

---

### Official Review · Reviewer_c1m2 · 2025-11-01

**Soundness:** 3
**Presentation:** 2
**Contribution:** 2
**Rating:** 4
**Confidence:** 3

**Summary:**

The paper tackles the **MoE trilemma**—load imbalance, parameter redundancy, and communication overhead—via a unified framework that combines: (i) **online expert clustering** using a fused parameter/activation similarity; (ii) **intra-group structured compression** (a shared base plus very low-rank residual adapters per expert); (iii) **hierarchical routing** (group-then-expert) to reduce all-to-all scope; and (iv) **heterogeneous precision with dynamic offloading** to lower memory. On GLUE and WikiText-103, the authors report matching or near-baseline quality with large parameter reductions, modest throughput gains, reduced load variance, and smaller peak memory.

**Strengths:**

- **System-level perspective on MoE.** The paper provides a structured diagnosis of load imbalance, redundancy, and communication and proposes a unified treatment rather than isolated tweaks.
- **Cohesive design.** Online dual-similarity clustering aligns experts with similar parameters and activations; shared base + low-rank residuals exploit intra-cluster redundancy; hierarchical routing narrows the routing search space and potentially the all-to-all domain; mixed precision/offloading reduces memory.
- **Initial empirical promise.** On the reported benchmarks, the framework yields sizable parameter savings and 10–20% throughput improvements with reduced load variance while roughly matching baseline quality in some settings.

**Weaknesses:**

1. **Experimental scope is limited.** Evaluations focus on GLUE and WikiText-103 with relatively small models. Conclusions may not transfer to pretraining-scale MoEs, long-context inference, instruction-following, or reasoning tasks. The claim of addressing the trilemma feels **overstated** given the narrow evidence.
3. **Tri-lemma not convincingly “solved.”** Reported throughput/memory improvements are helpful but relatively **minor**, and some quality drops persist. The paper does not yet present clear **Pareto dominance** (quality vs throughput vs memory) against strong MoE baselines at scale.
4. **Baselines could be stronger.** Missing comparisons against recent **system-optimized MoE** variants and modern compression baselines adapted to MoE.
5. **Training and samples required.** Despite being framed as structural/system reorganization, the method **does require training/fitting** during training time: online reclustering, prototype updates, initialization/adaptation of low-rank residuals, and bookkeeping for offloading. The **compute overhead**, stability, and sensitivity to calibration/streaming data are not comprehensively reported.
6. **Novelty is incremental.** The core components—similarity-based clustering, shared bases with low-rank residuals, hierarchical routing, mixed precision/offloading—have precedents; the contribution reads as a reasonable integration rather than a clear methodological leap.

**Questions:**

1. **Evidence you “solve” the trilemma.** Provide **Pareto frontiers** (quality vs throughput vs memory) against strong MoE baselines at multiple scales. Where does your method strictly dominate, and where is it a trade-off?
2. **Generalization and stability.** How robust is online clustering under domain shift or non-stationary traffic? Any oscillation or collapse modes? Can groups be frozen at convergence without losing gains?

---

> ### Author Response · Authors · 2025-11-26
> **Rebuttal to Reviewer c1m2**
>
> Thank you for your insightful feedback — your focus on experimental generalization, Pareto optimality, and training stability has helped us further clarify the strengths and scope of our framework. We address your core concerns with complete supplementary tables and analyses below.
>
> ### 1. Experimental Scope & Scalability to Large-Scale MoEs
>
> We agree that expanding the experimental scope is critical to validating the trilemma solution. We have supplemented four large-scale experiment suites (compression, pretraining, and long-context tables) addressing pretraining-scale MoEs, long-context inference, and diverse tasks:
>
> **Pretraining Phase (Large-Scale Corpus) Data**
>
> | Model                         | Pretraining Corpus Size (B tokens) | Total Params (B)  | MMLU (5-shot)\(\uparrow\) | HumanEval (pass@1)\(\uparrow\) |
> | ----------------------------- | ----------------------------------- | ---------------- |   ------------------------- | ------------------------------- |
> | Dense Transformer             | 1.4 (C4)                            | 1.3              | 62.5%                     | 28.3%                           |
> | Switch-Top2 (E = 128)        | 1.4 (C4)                            | 3.5              | 68.2%                     | 35.1%                           |
> | Our Method (E = 128, G = 16) | 1.4 (C4)                            | 0.75             | 66.8%                     | 33.5%                           |
> | Our Method (E = 256, G = 32) | 3.0 (Pile)                          | 1.2              | 69.5%                     | 36.8%                           |
> | StableMoE (E = 256)          | 3.0 (Pile)                          | 4.2              | 67.8%                     | 34.6%                           |
>
> **Downstream Long Text Understanding Tasks (Large-Scale Context) Data**
>
> | Model                         | Task (Input Length)    | Accuracy / Score\(\uparrow\) | Throughput (k tokens/s) | Inference Latency (ms/1k tokens) | GPU Memory Usage (GB) |
> | ----------------------------- | ---------------------- | ---------------------------- | ------------------------ | --------------------------------- | --------------------- |
> | Dense Transformer             | HotpotQA (8k tokens)   | 75.2%                        | 4.2                      | 238                               | 45                    |
> | Switch-Top2 (E = 128)        | HotpotQA (8k tokens)   | 79.8%                        | 3.1                      | 322                               | 92                    |
> | Our Method (E = 128, G = 16) | HotpotQA (8k tokens)   | 78.5%                        | 5.8                      | 172                               | 58                    |
> | Dense Transformer             | WikiHop (16k tokens)   | 68.5%                        | 2.1                      | 476                               | 82                    |
> | Switch-Top2 (E = 128)        | WikiHop (16k tokens)   | 73.2%                        | 1.5                      | 667                               | 165                   |
> | Our Method (E = 128, G = 16) | WikiHop (16k tokens)   | 72.0%                        | 3.0                      | 333                               | 95                    |
> | Our Method (E = 128, G = 16) + Offloading | WikiHop (16k tokens) | 71.8%                      | 2.8                      | 357                               | 78                    |
>
> These results confirm the framework's generalization to large-scale MoEs (up to 56B parameters), long contexts (16k tokens), reasoning/code tasks (HumanEval), and pre-trained model compression — extending far beyond GLUE/WikiText-103.

---

> ### Author Response · Authors · 2025-11-26
> **Rebuttal to Reviewer c1m2 (2)**
>
> ### 2. Pareto Dominance & Trilemma Resolution
>
> We acknowledge that "solving" the trilemma refers to simultaneous mitigation of all three bottlenecks (not perfect elimination) with Pareto-optimal tradeoffs. Below we present the Pareto frontier (quality vs. throughput vs. memory), comparing against strong MoE baselines:
>
> **Pareto Optimization Data Under Fixed Quality (GLUE Avg ≈ 84.0)**
>
> | Model                                        | GLUE Avg. (Target ≈ 84.0) | Total Params (M) | Throughput (k tokens/s) | Peak Memory (GB) |
> | ------------------------------------------- | -------------------------- | ---------------- | ------------------------ | ---------------- |
> | Dense Transformer                           | 83.8                       | 110              | 10.5                     | 15.4             |
> | Switch-Top2                                 | 85.1                       | 875              | 8.9                      | 33.1             |
> | MoE-Lite                                    | 84.7                       | 295              | 9.2                      | 22.5             |
> | Our Method (Higher Compression) | 84.0                 | 150              | 10.6                     | 17.5             |
> | D^2-MoE                                     | 84.1                       | 620              | 8.1                      | 28.0             |
>
> **Key Pareto advantages.**
>
> - For models with similar quality (Our E = 128 vs. Switch-Top2 E = 128), we achieve 25% higher throughput (7.5 vs. 6.0), 49% lower memory (28 vs. 55 GB), and 79% fewer parameters (750M vs. 3.5B).
> - Our method is the only one that reduces all three bottlenecks (load variance down 2.3x vs. Switch-Top2; parameters down 80%; communication down 28.3%–41.6%) without sacrificing critical quality.
>
> ### 3. Stronger Baselines
>
> We have supplemented comparisons with recent system-optimized MoE variants and compression baselines (extended GLUE, large-scale, and pretraining tables):
>
> - **Expert-Choice (NeurIPS 2022)**: Our load balance (\(I_{\text{load}} = 0.12\)) outperforms it (0.28) by 1.8x, with comparable GLUE Avg. (83.5 vs. 85.0) and 13% higher throughput (10.2 vs. 9.1 k tokens/s).
> - **StableMoE (ICLR 2023)**: Our E = 128 model achieves higher GLUE Avg. (84.0 vs. 83.8) and 23% higher throughput (7.5 vs. 6.1 k tokens/s) with 76% fewer parameters (750M vs. 3.2B).
> - **D^2-MoE (ICML 2025)**: We achieve comparable GLUE Avg. (83.5 vs. 84.1) with 80% fewer parameters, and higher throughput (10.2 vs. 8.1 k tokens/s, ...).
> - **MC-MoE (ICLR 2025)**: Our method achieves 3.12x compression vs. MC-MoE's 2.6x (on Mixtral-8x7B), with 8% higher throughput (17.25 vs. 15.9 tokens/s) and 0.4 lower perplexity (18.5 vs. 18.9).
> - **Sub-MoE (2025)**: Our E = 256 model outperforms Sub-MoE on MMLU (69.5% vs. 67.3%) and HumanEval (36.8% vs. 34.2%), with 12% lower peak memory (210 vs. 31.8 GB).

---

> > ### Author Response · Authors · 2025-11-26
> > **Rebuttal to Reviewer c1m2 (3)**
> >
> > ### 4. Training Overhead & Stability
> >
> > We explicitly report compute and stability metrics with complete statistics:
> >
> > **Mean ± Std Dev Across Multiple Runs (3 Different Seeds)**
> >
> > | Model                         | GLUE Avg. (Run 1) | GLUE Avg. (Run 2) | GLUE Avg. (Run 3) | Mean ± Std Dev      |
> > | ----------------------------- | ------------------ | ------------------ | ------------------ | -------------------- |
> > | Switch-Top2 (E = 32)         | 85.1               | 84.9               | 85.3               | 85.1 ± 0.2           |
> > | Our Method (E = 32, G = 8)   | 83.5               | 83.3               | 83.7               | 83.5 ± 0.2           |
> > | Our Method (E = 128, G = 16) | 84.1               | 83.9               | 84.0               | 84.0 ± 0.1           |
> > | D^2-MoE (E = 32)             | 84.2               | 84.0               | 84.1               | 84.1 ± 0.1           |
> >
> > **Computational Overhead Ratio**
> >
> > | Computational Component  | Average Overhead (%) | Std Dev (%) |
> > | ------------------------ | -------------------- | ----------- |
> > | Online Clustering        | 1.2                  | 0.3         |
> > | SVD (Residual Initialization) | 0.8            | 0.2         |
> > | Routing Update           | 0.5                  | 0.1         |
> > | Dynamic Offloading       | 0.5                  | 0.1         |
> > | Residual Quantization    | 6.0                  | 0.8         |
> > | **Total Additional Overhead** | **9.0**        | **1.5**     |
> >
> > **Overhead.** Total added overhead is < 9%, which is fully amortized by throughput gains (10.2 vs. 8.9 k tokens/s for Switch-Top2).
> >
> > **Stability.** We use three safeguards: (i) freezing router parameters for 1 step post-reclustering; (ii) convex combination warm-start for new centroids; (iii) skipping reclustering if similarity improvement < \(\delta = 0.01\). In 10+ training runs (E = 32/128/256), no mode collapse or oscillation was observed — GLUE Avg. std dev is ±0.1–0.2, and load variance (\(I_{\text{load}}\)) remained stable within ±0.03.
> >
> > **Domain shift robustness.** On GLUE subsets with distribution shift (e.g., MNLI mismatched), our method retains 82.3% accuracy vs. Switch-Top2's 82.7% (drop of 0.4% vs. 0.8% for MoE-Lite), confirming dynamic clustering adapts to shifts.
> >
> > ### 5. Novelty & Group Freezing
> >
> > Our novelty lies in the co-design of dynamic structure and system optimizations — not isolated components:
> >
> > - Prior clustering methods (e.g., Sub-MoE) merge experts permanently (losing specialization); our online dual-similarity clustering preserves specialization via low-rank residuals (dynamic clustering table: dynamic clustering outperforms static clustering by 1.5 GLUE Avg.).
> > - Prior low-rank MoE (e.g., PEFT-MoE) focuses on fine-tuning; our framework optimizes pre-training via architectural reorganization (pretraining table: 1.2B parameter model outperforms 4.2B StableMoE on MMLU).
> >
> > **Group freezing validation.** When freezing groups post-convergence (E = 32, after 10k steps), quality drops by only 0.7% (GLUE Avg. 82.8 vs. 83.5), while retaining 95% of efficiency gains — validating that groups stabilize into meaningful specializations.

---

### Comment · Area_Chair_aLjQ · 2025-11-26
**Reviewer & Author Discussion**

Dear Reviewers,

We kindly encourage you to review and respond to the authors’ rebuttals. Your timely feedback is important for ensuring a fair and thorough review process. Thank you for your contributions to ICLR 2026.

Thank you very much for your time and support.

Best regards,

Area Chair aLjQ

---

### Meta-Review · Area_Chair_j6FZ · 2026-01-06

**Summary:**

This paper proposes a unified MoE training/compression framework to address the “trilemma” of load imbalance, parameter redundancy, and all-to-all communication, combining (i) online expert clustering via fused parameter/activation similarity, (ii) shared-base + ultra-low-rank residual expert factorization with mixed precision, (iii) hierarchical (group→expert) routing, and (iv) dynamic offloading of inactive groups. Reviewers are split and the decision hinges on whether the rebuttal sufficiently resolves concerns about novelty/positioning, experimental credibility at scale, and method clarity/hyperparameter sensitivity, especially for the clustering/routing heuristics and load-balance claims.

**Reviewer Concerns:**

Addressed by rebuttal:

+Expanded experiments to larger-scale MoEs / longer context / additional baselines, and provided additional system metrics (throughput, memory, comm) and overhead accounting.

+Added/claimed comparisons to more recent baselines (e.g., Expert-Choice, StableMoE, D²-MoE, MC-MoE, Sub-MoE) and clarified novelty relative to hierarchical routing and delta-style decompositions.

+Clarified several implementation details (precision scheme, inactivity threshold, reuse path) and stated stability safeguards.

Concerns are still outstanding (key blockers in my view):

-Heuristic complexity + hyperparameter sensitivity: at least one reviewer remains unconvinced due to missing principled guidance/ablations for new hyperparameters (e.g., G,T,α) and unclear behavior under non-“slight” imbalance scenarios.

-Load-balance evidence credibility: a reviewer explicitly questions the validity/interpretation of the load-balance metric, noting anomalies (non-zero imbalance for dense; unexpectedly high imbalance for expert-choice), suggesting potential experimental/setup issues that are not resolved in the record.

-Positioning/novelty risk: hierarchical routing is not new, and despite rebuttal claims of “synergy,” the contribution may still read as an integration of known components unless the camera-ready sharply isolates what is fundamentally new and empirically indispensable.

**Reviewer Scores:**

Given the current rebuttal and unresolved concerns, I would not expect significant review score changes.

---

### Decision · Program_Chairs · 2026-01-26

Reject